# ON VARIATIONAL LEARNING OF CONTROLLABLE REPRESENTATIONS FOR TEXT WITHOUT SUPERVISION

## ABSTRACT

The variational autoencoder (VAE) has found success in modelling the manifold of natural images on certain datasets, allowing meaningful images to be generated while interpolating or extrapolating in the latent code space, but it is unclear whether similar capabilities are feasible for text considering its discrete nature. In this work, we investigate the reason why unsupervised learning of controllable representations fails for text. We find that traditional sequence VAEs can learn disentangled representations through their latent codes to some extent, but they often fail to properly decode when the latent factor is being manipulated, because the manipulated codes often land in holes or vacant regions in the aggregated posterior latent space, which the decoding network is not trained to process. Both as a validation of the explanation and as a fix to the problem, we propose to constrain the posterior mean to a learned probability simplex, and performs manipulation within this simplex. Our proposed method mitigates the latent vacancy problem and achieves the first success in unsupervised learning of controllable representations for text. Empirically, our method significantly outperforms unsupervised baselines and is competitive with strong supervised approaches on text style transfer. Furthermore, when switching the latent factor (*e.g.*, topic) during a long sentence generation, our proposed framework can often complete the sentence in a seemingly natural way – a capability that has never been attempted by previous methods.

## 1 INTRODUCTION

High-dimensional data, such as images and text, are often causally generated through the interaction of many complex factors, such as lighting and pose in images or style and content in texts. Recently, VAEs and other unsupervised generative models have found successes in modelling the manifold of natural images (Higgins et al., 2017; Kumar et al., 2017; Chen et al., 2016). These models often discover controllable latent factors that allow manipulation of the images through conditional generation from interpolated or extrapolated latent codes, often with impressive quality. On the other hand, while various attributes of text such as sentiment and topic can be discovered in an unsupervised way, manipulating the text by changing these learned factors have not been possible with unsupervised generative models to the best of our knowledge. Cífka et al. (2018); Zhao et al. (2018) observed that text manipulation is generally more challenging compared to images, and the successes of these models cannot be directly transferred to texts.

Controllable text generation aims at generating realistic text with control over various attributes including sentiment, topic and other high-level properties. Besides being a scientific curiosity, the possibility of unsupervised controllable text generation could help in a wide range of application, *e.g.*, dialogues systems (Wen et al., 2016). Existing promising progress (Shen et al., 2017; Fu et al., 2018; Li et al., 2018; Sudhakar et al., 2019) all relies on supervised learning from annotated attributes to generate the text in a controllable fashion. The high cost of labelling large training corpora with attributes of interest limits the usage of these models, as pre-existing annotations often do not align with some downstream goal. Even if cheap labels are available, for example, review scores as a proxy for sentiment, the control is limited to the variation defined by the attributes.

In this work, we examine the obstacles that prevent sequence VAEs from performing well in unsupervised controllable text generation. We empirically discover that manipulating the latent factors for typical semantic variations often leads to latent codes that reside in some low-density region of the

aggregated posterior distribution. In other words, there are *vacant* regions in the latent code space (Makhzani et al., 2015; Rezende & Viola, 2018) not being considered by the decoding network, at least not at convergence. As a result, the decoding network is unable to process such manipulated latent codes, yielding unpredictable generation results of low quality.

In order to mitigate the latent vacancy problem, we propose to constrain the posterior mean to a learned probability simplex and only perform manipulation within the probability simplex. Two regularizers are added to the original objective of VAE. The first enforces an orthogonal structure of the learned probability simplex; the other encourages this simplex to be filled without holes. Besides confirming that latent vacancy is indeed a cause of failure in previous sequence VAEs', it is also the first successful attempt towards unsupervised learning of controllable representations for text to the best of our knowledge. Experimental results on text style transfer show that our approach significantly outperforms unsupervised baselines, and is competitive with strong supervised approaches across a wide range of evaluation metrics. Our proposed framework also enables finer-grained and more flexible control over text generation. In particular, we can switch the topic in the middle of sentence generation, and the model will often still find a way to complete the sentence in a natural way.

## 2 BACKGROUND: VARIATIONAL AUTOENCODERS

The variational autoencoder (VAE) (Kingma & Welling, 2013) is a generative model defined by a prior $p(\boldsymbol{z})$ and a conditional distribution $p_{\boldsymbol{\theta}}(\boldsymbol{x}|\boldsymbol{z})$. The VAE is trained to optimize a tractable variational lower bound of $\log p_{\boldsymbol{\theta}}(\boldsymbol{x})$:

$$\mathcal{L}_{\text{VAE}}(\boldsymbol{x}; \boldsymbol{\theta}, \boldsymbol{\phi}) = \mathbf{E}_{\boldsymbol{z} \sim q_{\boldsymbol{\phi}}(\boldsymbol{z}|\boldsymbol{x})}[\log p_{\boldsymbol{\theta}}(\boldsymbol{x}|\boldsymbol{z})] - D_{\text{KL}}(q_{\boldsymbol{\phi}}(\boldsymbol{z}|\boldsymbol{x})||p(\boldsymbol{z})), \tag{1}$$

where $q_{\boldsymbol{\phi}}(\boldsymbol{z}|\boldsymbol{x})$ is a variational distribution parameterized by an encoding network with parameters $\boldsymbol{\phi}$, and $p_{\boldsymbol{\theta}}(\boldsymbol{x}|\boldsymbol{z})$ denotes the decoding network with parameters $\boldsymbol{\theta}$. This objective tries to minimize the reconstruction error to generate the data, and at the same time regularizes $q_{\boldsymbol{\phi}}(\boldsymbol{z}|\boldsymbol{x})$ towards the prior $p(\boldsymbol{z})$. In this paper, $p(\boldsymbol{z})$ is chosen as $\mathcal{N}(\boldsymbol{0}, \boldsymbol{I})$. For text modelling, the input $\boldsymbol{x}$ is some observed text. Both the encoding and decoding network are usually recurrent neural networks.

Note that during learning, the decoding network $p_{\boldsymbol{\theta}}$ only learns to decode conditioned on $\boldsymbol{z}$ that are sampled from $q_{\boldsymbol{\phi}}(\boldsymbol{z}|\boldsymbol{x})$. In other words, the decoding network only learns to process $\boldsymbol{z}$ sampled from the aggregated posterior distribution $q_{\boldsymbol{\phi}}(\boldsymbol{z}) = \mathbf{E}_{\boldsymbol{x} \sim p_d(\boldsymbol{x})} q_{\boldsymbol{\phi}}(\boldsymbol{z}|\boldsymbol{x})$, where $p_d(\boldsymbol{x})$ is the data distribution. If $q_{\boldsymbol{\phi}}(\boldsymbol{z})$ has regions of low density, there is no guarantee that $p_{\boldsymbol{\theta}}$ would decode well in such regions. This is an important intuition that will become central to our analysis in Sec. 3.

## 3 LATENT VACANCY PREVENTS EFFECTIVE MANIPULATION

In this section, we take a deeper look into the aggregated posterior latent space of sequence VAE trained on text, and provide justification for the alternative solution we propose in Section 4.

### 3.1 OBSERVATIONS FROM UNSUPERVISED SENTIMENT MANIPULATION

As pointed out by Bowman et al. (2015), one of the motivations to apply VAEs on text is to allow generation of the sentences conditioned on extrinsic features by controlling the latent codes. Without annotated labels, no previous methods have successfully learned controllable latent factors as mentioned in Sec. 1. To understand what is missing, we conduct exploratory experiments to use VAE for unsupervised sentiment manipulation.

We use the Yelp restaurant reviews dataset and the same data split following Li et al. (2018). We train a $\beta$-VAE (Higgins et al., 2017)[1] with a latent space of 80 dimensions, an LSTM encoder, and an LSTM decoder. Details about this experiment are described in Appendix A.1.

By inspecting the accuracy on the validation set, we find that there exists one dimension of latent code achieving higher than $90\%$ sentiment classification accuracy by its value alone, while other latent codes get accuracy around $50\%$. Further details can be found in Appendix A.2. It means that this latent dimension is an effective sentiment indicator. Similar phenomena have been observed in

---

[1]We also try state-of-the-art techniques (He et al., 2019) on VAE w.r.t. optimizing ELBO but the KL term trained with those techniques are too small to capture the details of the source sentence.

large-scale language models (Radford et al., 2017). However, the direct influence on the generative process of the model observed in Radford et al. (2017) does not apply on the VAE. When we try to perform sentiment manipulation by modifying this latent dimension[2], the decoding network fails to generate the desired outputs most of the time, as evidenced by the poor quantitative evaluation in Table. 1, and poor samples shown in Appendix A.3.

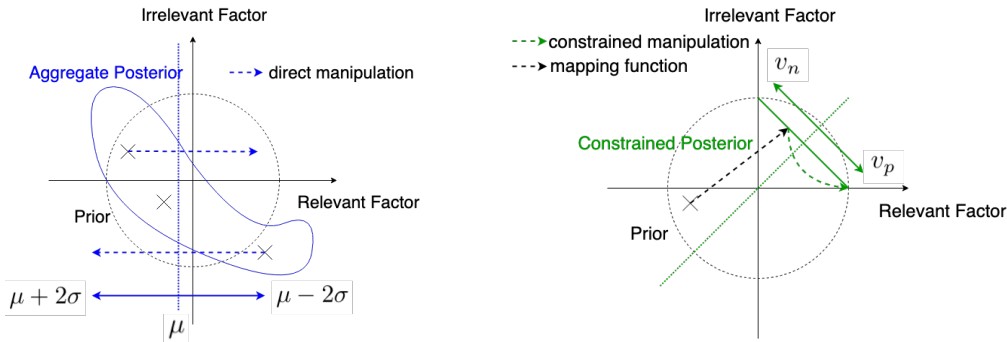

Figure 1: Illustration of why latent vacancy prevents effective manipulation in VAEs.

Figure 2: CP-VAE, mapping the posterior to a probability simplex with orthogonal basis vectors.

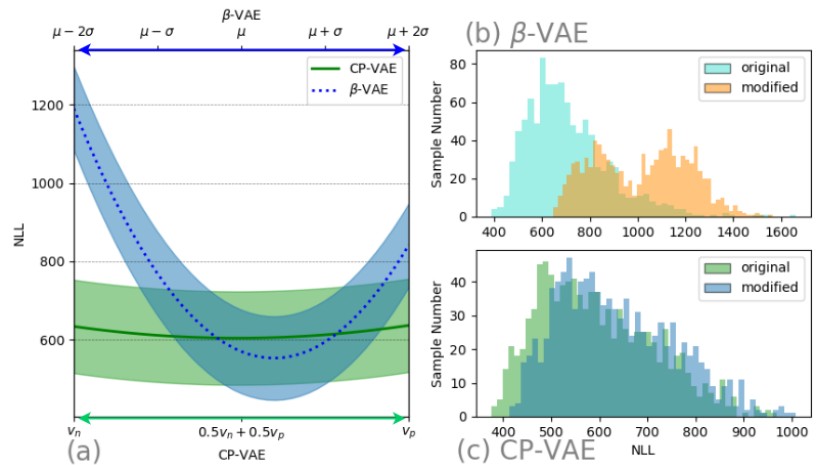

Figure 3: (a) Comparisons between $\beta$-VAE and CP-VAE considering density under the aggregated posterior distribution, the blue and green solid arrows match with the manipulation illustrated in Fig 1 and 2; (b) Histogram of original and modified latent codes' NLL in $\beta$-VAE; (c) Histogram of original and modified latent codes' NLL in CP-VAE.

## 3.2 LATENT VACANCY IN TEXT MODELLING

One possible reason for the failure is that the decoding network is never trained on codes like the manipulated ones. This is the case if the aggregated posterior has holes or regions of low density, and the manipulated codes fall into such vacant regions. Supposing the aggregated posterior latent space possesses a shape as shown in Fig. 1, the direct manipulated latent codes will fall out of the aggregated posterior latent space for most input samples. Such latent codes are never seen by the model during training and possess a low density under the aggregated posterior distribution, leading to unpredictable behaviours during decoding.

To verify our hypothesis demonstrated in Fig. 1, we empirically estimate the density of sentiment-manipulated codes under the aggregated posterior distribution of our trained VAE. Here, we approximate the data distribution $p_d(\boldsymbol{x})$ with the empirical distribution over all the training samples. As a result, the estimated aggregated posterior distribution is a large mixture of Gaussian distribution. For all 1000 test samples, we move the dimension of code capturing sentiment from $\mu - 2\sigma$ to $\mu + 2\sigma$ where $\mu$ and $\sigma$ are the mean and the standard deviation estimated on all the training samples and

---

[2]Different strategies are attempted, see Appendix A.4 for details.

measure the averaged negative log-likelihood (NLL) under the aggregated posterior distribution. As depicted in Fig. 3 (a), the NLL plotted in blue dot curve rises sharply when moving away from $\mu$ even if there is only one dimension of code is changing, indicating the existence of the vacancy in the aggregated posterior latent space. In addition, we draw the histogram of all the test samples' NLL considering their original latent codes and modified ones in Fig. 3 (b). The histogram shows that there is a large divergence in NLL between the original latent codes and the modified ones. Also, the modified latent codes have two separate modes, confirming the irregular shape of the aggregated posterior latent space. In order to resolve this issue, the approach proposed in this work is to constrain the posterior in a way that the manipulation only happens in a learned simplex, as depicted in Fig. 2. In this constrained subspace, the phenomenom of low density holes of aggregated posterior is significantly reduced, as Fig. 3 (a) and (c) empirically show that there is little change in NLL of original versus modified codes. The details of our method is presented in the next section.

## 4 METHOD

### 4.1 OVERVIEW

The experiments conducted in Sec. 3 validates the existence of vacancy in the aggregated posterior latent space. One potential way to resolve the problem is to better match the aggregated posterior with the prior (Makhzani et al., 2015; Tomczak & Welling, 2017; Zhao et al., 2018). However, in terms of unsupervised learning of controllable representation for text, these previous methods have not shown successes; Zhao et al. (2018) only attempted supervised text style transfer, and also reported negative results from the AAE (Makhzani et al., 2015). Another way to resolve the vacancy issue is to directly enforce that the aggregated posterior itself has no vacant region anywhere where we would like to perform latent code manipulation. We propose to map the posterior Gaussian mean to a constrained space, more specifically a learned probability simplex, where we can encourage the constrained latent space to be filled without vacancy, and perform manipulation to be within this simplex. As illustrated in Fig. 2, we add an additional mapping function as part of the encoding network which maps the mean of the Gaussian posterior to a constrained space. Two regularization terms are introduced later to ensure the learned simplex is not degenerate and that this subspace is well filled.

In addition, we separately model the relevant factors that we wish to control and the irrelevant factors by splitting $\boldsymbol{z}$ into two parts, $\boldsymbol{z}^{(1)}$ and $\boldsymbol{z}^{(2)}$, following prior work (Bao et al., 2019). The first part captures the relevant factors that are dominant in the data without an inductive bias from external signals, while the second part learns to encode the remaining local information that is useful for reconstructing the source sentences. As a result, $q_{\boldsymbol{\phi}}(\boldsymbol{z}|\boldsymbol{x})$ is decomposed into $q_{\boldsymbol{\phi}_1}(\boldsymbol{z}^{(1)}|\boldsymbol{x})q_{\boldsymbol{\phi}_2}(\boldsymbol{z}^{(2)}|\boldsymbol{x})$ where $\boldsymbol{\phi} = \boldsymbol{\phi}_1 \cup \boldsymbol{\phi}_2$. With diagonal covariances the KL divergence term in Eq. 1 splits into two separate KL terms. In practice, we use a MLP encoding network to parametrize $\boldsymbol{z}^{(1)}$ with some sentence representations as the input (*e.g.*, averaging GloVe embeddings (Pennington et al., 2014) over the input tokens) and a LSTM encoding network to parametrize $\boldsymbol{z}^{(2)}$. We only constrain the posterior of $\boldsymbol{z}^{(1)}$ and $\boldsymbol{z}^{(2)}$ is optimized the same way as the traditional VAE.

### 4.2 CONSTRAINING THE POSTERIOR

We now describe how to map the mean $\boldsymbol{\mu}$ of the Gaussian posterior for $\boldsymbol{z}^{(1)} \in \mathbb{R}^N$ to a constrained latent space. We would like to constrain the mean $\boldsymbol{\mu}$ to have a structure as follows:

$$\boldsymbol{\mu} = \sum_{i=1}^{K} p_i \boldsymbol{e}_i, \quad \sum_{i=1}^{K} p_i = 1, \quad \langle \boldsymbol{e}_i, \boldsymbol{e}_j \rangle = 0, i \neq j, \quad K \leq N \tag{2}$$

where $\boldsymbol{e}_i$ are vectors representing the relevant factors, $p_i$ is the proportion of $i$th relevant factor encoded in $\boldsymbol{z}^{(1)}$ and $K$ is a hyperparameter indicating the number of relevant factors to discover. In other words, the mean of the Gaussian posterior of $\boldsymbol{z}^{(1)}$ is constrained to be inside a $K$-dimension probability simplex in $\mathbb{R}^N$ whose vertices are represented by the orthogonal basis vectors $\boldsymbol{e}_i, i = 1, \ldots, K$. Given the outputs of the MLP encoder $\boldsymbol{h}$ and $\log \boldsymbol{\sigma}^2$, we learn an additional mapping function $\pi$ which maps $\boldsymbol{h}$ to the constrained posterior space, which can be treated as part of the encoding network:

$$\boldsymbol{\mu} = \pi(\boldsymbol{h}) = \boldsymbol{E} \cdot \text{softmax}(\boldsymbol{W}\boldsymbol{h} + \boldsymbol{b}), \tag{3}$$

where $\boldsymbol{E} = [\boldsymbol{e}_1, \ldots, \boldsymbol{e}_K]$ is a learnable embedding matrix representing the bases, $\boldsymbol{W}$ is the learnable weight matrix, and $\boldsymbol{b}$ is the learnable bias vector. As a result, the constrained posterior is parametrized by $\boldsymbol{\mu}$ and $\log \boldsymbol{\sigma}^2$ as a Gaussian distribution $\mathcal{N}(\boldsymbol{\mu}, \text{diag}(\boldsymbol{\sigma}^2))$.

With the mapping function alone, the proposed VAE suffers from posterior collapse (Bowman et al., 2015), a well-known problem where the model ignores the latent code $z$ during the training. Further complicating matters is the fact that there is an abundance of signals for predicting the next token in the text, but the signals indicating high-level semantics are quite sparse. It is thus unlikely that the VAEs can capture useful relevant factors from raw text without collapse. For these reasons, we enforce orthogonality in the learnt basis vectors as defined in Eq. 2, which introduces a natural recipe to prevent posterior collapse for $z^{(1)}$. Note that the KL divergence between $q_{\phi_1}(z^{(1)}|x)$ and $p(z^{(1)})$ is

$$D_{\mathrm{KL}}(q_{\phi_1}(z^{(1)}|x)\|p(z^{(1)})) = \frac{1}{2}\mu^\top\mu + \frac{1}{2}\left(\sigma^\top\sigma - \log\sigma^\top\sigma - 1\right). \tag{4}$$

With orthogonality in the basis vectors, the first term in the above equation can be factorized into

$$\mu^\top\mu = (\sum_i p_i e_i)^\top(\sum_i p_i e_i) = \sum_i p_i^2 e_i^\top e_i. \tag{5}$$

To encourage orthogonality in the basis vectors, a regularization term is added to the objective function:

$$\mathcal{L}_{\mathrm{REG}}(x;\phi_1) = \|E^\top E - \alpha I\|, \tag{6}$$

where $I$ is the identity matrix and $\alpha$ is a hyperparamter. When $\mathcal{L}_{\mathrm{REG}} = 0$, $e_i^\top e_i = \alpha$. In this case, $\mu^\top\mu = \alpha\sum_i p_i^2$ reaches its minimum $\frac{\alpha}{K}$ when $p$ is a uniform distribution. The proof can be found in Appendix C. In practice, $\mathcal{L}_{\mathrm{REG}}$ will quickly decrease to around 0, ensuring that the KL term will never fully collapse with the structural constraint. When it comes to controlled generation, one can choose a vertex or any desired point in the probability simplex, as illustrated in Fig. 2.

Note that the constrained posterior also means that the aggregated posterior can never match the isotropic Gaussian prior. In other word, we achieve good controlled text generation potentially at the cost of poor uncontrolled generation from the prior, but such is not the focus of this current work, and could potentially be resolved by selecting or learning a better prior as in Tomczak & Welling (2017).

### 4.3 FILLING THE CONSTRAINED SPACE

Constraining the posterior inside a certain space does not guarantee that this space will be filled after training. In order to prevent this, we want the probability distribution over the relevant factors $p$ to cover as much of the constrained latent space as possible. We introduce a reconstruction error of the structured latent code in order to push $p$ away from a uniform distribution. For each input sentence, we randomly sample $m$ sentences from the training data as negative samples. By applying the same encoding process, we get the structured latent code $\mu_i^{(-)}$ for each negative sample. Our goal is to make the raw latent code $h$ similar to the restructured latent code $\mu$ while different from latent codes $\mu_i^{(-)}$ of the negative samples, so that $p$ is generally different for each input sample. The structured reconstruction loss is formulated as a margin loss as follows:

$$\mathcal{L}_{\mathrm{S\text{-}REC}}(x;\phi_1) = \mathbb{E}_{z^{(1)}\sim q_{\phi_1}(z^{(1)}|x)}\left[\frac{1}{m}\sum_{i=1}^m \max(0, 1 - h\cdot\mu + h\cdot\mu_i^{(-)})\right]. \tag{7}$$

Our final objective function is defined as follows:

$$\mathcal{L}(x;\theta,\phi) = \mathcal{L}_{\mathrm{VAE}} + \mathcal{L}_{\mathrm{REG}} + \mathcal{L}_{\mathrm{S\text{-}REC}}. \tag{8}$$

## 5 RELATED WORK

### 5.1 UNSUPERVISED LEARNING OF DISENTANGLED REPRESENTATIONS

Learning disentangled representations is an important step towards better representation learning (Bengio et al., 2013) which can be useful for (semi-)supervised learning of downstream tasks, transfer and few-shot learning (Peters et al., 2017). VAEs have achieved promising results for unsupervised learning of disentangled representations. Several variations of VAEs have been proposed to achieve better disentanglement (Higgins et al., 2017; Kumar et al., 2017; Chen et al., 2016; Razavi et al., 2019). However, most recent progress in this direction has been restricted to the domain of images.

## 5.2 CONTROLLED TEXT GENERATION

In order to perform controllable text generation, previous methods either assume annotated attributes or multiple text datasets with different known styles (Hu et al., 2017; Shen et al., 2017; Zhao et al., 2018; Fu et al., 2018; Li et al., 2018; Sudhakar et al., 2019; Logeswaran et al., 2018; Lample et al., 2018). The requirement of labelled data largely restricts the capabilities and the applications of these models. Instead, all our proposed framework needs is raw text without any annotated attribute. The dominant underlying relevant factors in the given corpus will be discovered and disentangled by our unsupervised method, which can in turn be used for controlled generation.

## 6 EXPERIMENTS

To demonstrate the effectiveness of our approach, we compare it to unsupervised baselines with traditional VAEs, considering the density under the aggregated posterior distribution and the performance on sentiment manipulation. Following evaluation protocols in text style transfer, we also compare our method to strong supervised approaches. Furthermore, we showcase the ability of finer-grained style discovery and transition possessed by our system, which has not been attempted in the literature.

In this section, our proposed framework is referred as CP-VAE (Constrained Posterior VAE). Detailed configurations including the hyperparameters, model architecture, training regimes, and decoding strategy are found in Appendix B.

### 6.1 COMPARISONS WITH UNSUPERVISED BASELINES

**Experimental setup:** We use the same experimental setting and dataset as mentioned in Sec. 3. The 80D latent code is split into 16 and 64 dimensions for $z^{(1)}$ and $z^{(2)}$ respectively. The sentence representations used for $z^{(1)}$ is the averaged GloVe embeddings over the input tokens and $K$ is chosen as 3. To decide which basis vector corresponds to which sentiment, we sample 10 positive and 10 negative sentences respectively in the development set, pass them to the encoder, and choose the basis vector with the highest average $p_i$ in $\boldsymbol{p} = \text{softmax}(\boldsymbol{Wh} + \boldsymbol{b})$, yielding $v_p$ as the positive basis and $v_n$ as the negative basis. If $v_p$ and $v_n$ are chosen to be the same vector, we choose the index with the second highest $p_i$ for $v_p$. To perform sentiment manipulation, we fix $z^{(1)}$ to be the chosen basis vector; that is, $v_p$ or $v_n$.

**Comparisons on density under the aggregated posterior distribution:** First, we do linear interpolation between the two discovered basis vectors $v_p$ and $v_n$ and estimate the averaged NLL under the aggregated posterior distribution the same way as introduced in Sec. 3. The green solid curve in Fig. 3 (a) shows that the NLL of CP-VAE is relatively stable for the whole range of the interpolation. In Fig. 3 (c), the original latent codes and the modified ones largely overlap with each other. Both observations validate the effectiveness of CP-VAE in resolving the latent vacancy problem, leading to significant improvements on unsupervised sentiment manipulation, as seen later.

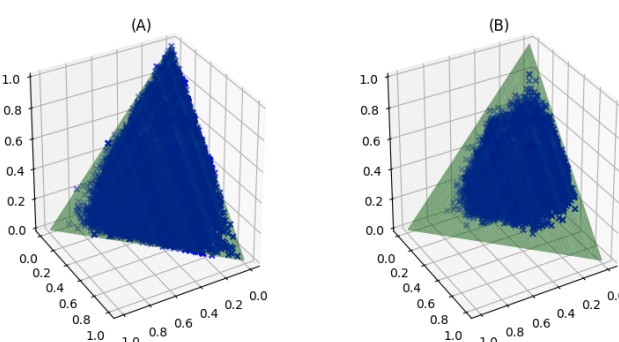

Figure 4: Visualization of all training samples in the probability simplex: (A) With $\mathcal{L}_{\text{S-REC}}$ ;(B) Without $\mathcal{L}_{\text{S-REC}}$.

**Comparsions with metrics on text style transfer:** For quantitative evaluation, we adopt automatic evaluation metrics used in text style transfer (Sudhakar et al., 2019) including classification accuracy

(AC), BLEU score (BL), GLEU score (GL) and language model perplexity (PL), whose definitions are elaborated in the next section. We also report $D_{\text{KL}}(q_{\phi_1}(z^{(1)}|x)\|p(z))$ (KL) for $z^{(1)}$ of CP-VAE. As shown in Tab. 1, CP-VAE performs significantly better than $\beta$-VAE in terms of accuracy, BLEU and GLEU. The lower perplexity of $\beta$-VAE is due to mode collapse, which produces very short pivot sentences such as "great !". The results match our observations from the experiments on density under the aggregated posterior distribution, confirming that latent vacancy prevents effective manipulation of the latent codes. We also conduct an ablation study by removing $\mathcal{L}_{\text{REG}}$ and $\mathcal{L}_{\text{S-REC}}$ from the objective. The results demonstrate that both terms are crucial to the success of CP-VAE. Without $\mathcal{L}_{\text{REG}}$, CP-VAE experiences posterior collapse for $z^{(1)}$. As a result, $v_p$ and $v_n$ collide with each other, leading to failure in disentangled representation learning. Since we choose $K$ as 3, it is convenient to visualize the samples during training with $p$ in the learnt probability simplex, as shown in Fig. 4. We can see that the whole simplex is mostly covered with samples with the help of $\mathcal{L}_{\text{S-REC}}$. Without $\mathcal{L}_{\text{S-REC}}$, the decoding network fails to recognize the basis vectors due to the poor coverage of the probability simplex, causing the model to lose most of its transferring ability.

Table 1: Comparisons with unsupervised baselines on Yelp dataset.

| Model | AC $\uparrow$ | BL $\uparrow$ | GL $\uparrow$ | PL $\downarrow$ | KL for $z^{(1)}$ |
|---|---|---|---|---|---|
| $\beta$-VAE | $50.44 \pm 2.04$ | $7.68 \pm 0.33$ | $2.97 \pm 0.14$ | $\mathbf{24.63 \pm 1.85}$ | $NA$ |
| CP-G(loVe) | $\mathbf{60.22 \pm 4.57}$ | $33.69 \pm 1.47$ | $\mathbf{6.78 \pm 0.44}$ | $63.12 \pm 2.41$ | $18.35 \pm 0.15$ |
| - $\mathcal{L}_{\text{REG}}$ | $10.82 \pm 0.91$ | $33.27 \pm 2.84$ | $5.04 \pm 0.27$ | $37.25 \pm 2.40$ | $0.04 \pm 0.01$ |
| - $\mathcal{L}_{\text{S-REC}}$ | $12.28 \pm 3.69$ | $\mathbf{49.34 \pm 2.65}$ | $6.22 \pm 0.40$ | $55.26 \pm 2.46$ | $17.57 \pm 0.12$ |

## 6.2 COMPARISONS TO SUPERVISED APPROACHES ON TEXT STYLE TRANSFER

**Experimental setup:** We choose two datasets, Yelp and Amazon, used in works (Li et al., 2018; Sudhakar et al., 2019) on text style transfer which provide human gold-standard references for the test set. The same train-dev-test splits are used in our experiments. Two different sentence representations are used in this experiment, averaged GloVe and BERT (Devlin et al., 2018), denoted as **CP-G(loVe)** and **CP-B(ert)** respectively. The remaining settings are as described in the above section.

**Compared supervised approaches:** On the two datasets, we compare to three adversarially trained models: StyleEmbedding (**SE**) (Fu et al., 2018), MultiDecoder (**MD**) (Fu et al., 2018), CrossAligned (**CA**) (Shen et al., 2017) and two state-of-the-art models based on a "delete, transform, and generate" framework: DeleteAndRetrieve (**D&R**) (Li et al., 2018) and Blind-GenerativeStyleTransformer (**B-GST**) (Sudhakar et al., 2019).

**Evaluation protocols:** Four different automatic evaluation metrics are used to measure the different perspectives of the transferring quality, following Sudhakar et al. (2019). To measure transferring ability, we use pre-trained CNN based classifiers achieving 98% and 84% accuracies on the test sets of Yelp and Amazon respectively. To measure content preservation, we use the BLEU (Papineni et al., 2002) score between the transferred sentences and the source sentences. To measure fluency, we finetune OpenAI GPT-2 (Radford et al., 2019) with 345 million parameters on the same training-dev-test split to obtain the perplexity of generated sentences. The fine-tuned language models achieve perplexities of 26.6 and 34.5 on the test sets of Yelp and Amazon respectively. In addition, Sudhakar et al. (2019) argued that the Generalized Language Evaluation Understanding Metric (GLEU) has a better correlation with the human judgement. Here, we use the implementation of GLEU[3] provided by Napoles et al. (2015) to calculate the GLEU score.

**Result Analysis:** As observed by Li et al. (2018) and Sudhakar et al. (2019), accuracy, BLEU score and perplexity do not correlate well with human evaluations. Therefore, it is important to not consider them in isolation. Tab. 2 shows that our proposed approaches get similar scores on these metrics with human reference sentences on the second row, indicating that the generated sentences of our proposed approaches is reasonable considering the combination of these metrics. As seen by Sudhakar et al. (2019) and verified in Sec. 6.1, GLEU strike a balance between target style match and content retention and correlate well with the human evaluations. From Tab. 2, CP-VAE consistently outperforms the three adversarially trained models on GLEU by a noticeable margin and achieve competitive results as compared to the recent state-of-the-art models. By checking the samples generated from the models as shown in Tab. 3, B-GST, the current state-of-the-art, is more

---

[3]https://github.com/cnap/gec-ranking

Table 2: Comparisons with supervised approaches on Yelp and Amazon dataset.

| Model | Yelp | | | | Amazon | | | |
|---|---|---|---|---|---|---|---|---|
| | AC ↑ | BL ↑ | GL ↑ | PL ↓ | AC ↑ | BL ↑ | GL ↑ | PL ↓ |
| Source | 1.8 | 100.0 | 8.4 | 26.6 | 16.3 | 100.0 | 22.8 | 34.5 |
| Human | 70.1 | 25.3 | 100.0 | 63.7 | 41.2 | 45.7 | 100.0 | 68.6 |
| CA | 74.0 | 20.7 | 6.0 | 103.6 | **75.5** | 0.0 | 0.0 | **39.3** |
| SE | 8.2 | **67.4** | 6.9 | 65.4 | 40.2 | 0.4 | 0.0 | 125.0 |
| MD | 49.5 | 40.1 | 6.6 | 164.1 | 70.1 | 0.3 | 0.0 | 138.8 |
| D&R | **88.1** | 36.7 | 7.9 | 85.5 | 49.2 | 0.6 | 0.0 | 46.3 |
| B-GST | 85.6 | 45.2 | **12.7** | 49.6 | 55.2 | **52.3** | **18.1** | 48.2 |
| **CP-G** | 66.7 | 35.5 | 7.5 | 67.8 | 60.1 | 35.4 | 11.5 | 109.1 |
| **CP-B** | 55.4 | 48.4 | 9.6 | **47.6** | 40.0 | 39.7 | 12.7 | 97.3 |

Table 3: Samples of generated sentences. SRC is the input sentence and HUMAN is the human references.

| **Yelp** | *Positive to Negative* | *Negative to Positive* |
|---|---|---|
| SRC | this place is super yummy ! | but it probably sucks too ! |
| HUMAN | this place is super yucky ! | but it probably doesn't suck too ! |
| B-GST | this place is super bad ! | but it tastes great too ! |
| **CP-G** | this place is super slow and watered down . | but it 's truly fun and insanely delicious . |
| **CP-B** | this place is super greasy and gross ! | but it 's probably wonderful when you ! |

| **Amazon** | *Positive to Negative* | *Negative to Positive* |
|---|---|---|
| SRC | because it s made of cast iron , scorching is minimized . | they are cheerios, afterall, and we love the original kind . |
| HUMAN | because it is made of cast iron, scorching is maximized . | they are cheerios, and we love them . |
| B-GST | because it s cheaply made of cast iron , is useless . | they are sturdy, afterall, sturdy and we love the original . |
| **CP-G** | because it s made of cast iron , vomitting . | they are ripe, tastier , and we love them . |
| **CP-B** | because it s made of cast iron , limp . | they are divine, fluffier , and we love them . |

consistent to the source sentence, which can be expected, since it only makes necessary edits to flip the sentiment. CP-VAE tends to generate more diverse contents which may not be relevant sometimes, but the overall quality is reasonable considering it is trained without the label information. More samples can be found in Appendix E.

## 6.3 FINER-GRAINED STYLE DISCOVERY AND TRANSITION

Table 4: Two pairs of samples generated without and with topic transition. The first sentence in the pair is generated with a topic fixed throughout the generation; while the second sentence is generated with topic transition, the generated outputs after switching are marked as bold.

| *World* throughout | A federal judge on Friday ordered a federal appeals court to overturn a federal appeals court ruling that the Visa and MasterCard credit card associations violated federal antitrust law by barring the names of the state . |
|---|---|
| *World* to *Sci/Tech* | A federal judge on Friday ordered a federal appeals court to overturn a decision by the Supreme Court to **overturn a decision by the Federal Communications Commission to block the company's antitrust case against Microsoft Corp .** |
| *Sports* throughout | NEW YORK (Reuters) - Roger Federer, the world's No. 1 player, will miss the rest of the season because of a sore quadriceps . |
| *Sports* to *Business* | NEW YORK (Reuters) - Roger Federer, the world's No. 1 player, will miss the rest of the **year because of a bid-rigging scandal .** |

To further explore the potential of CP-VAE, we conduct the following exploratory experiments. We use the AG news dataset constructed by (Zhang et al., 2015), which contains four topic categories which are *World, Sports, Business* and *Sci/Tech*, with the title and description fields. Here, we drop the title and just use the description field to train CP-VAE and set $K = 10$. All four topics are automatically discovered by CP-VAE and identified as described in Sec. 6.1. We also compare the results of our identified topics to standard baselines for unsupervised topic modelling, the details

can be found in Appendix D. We choose a basis vector discovered by our model and generate a few tokens. Then, we switch the basis vector and continue the generation until the *end-of-seq* token is generated. Generated samples are shown in Table 4. We see that our model learns to transition from one topic to another in a natural and fluent way within the same sentence. Several observations can be made based on these samples: (1) it is good at detecting name entities and replacing them with the name entities related to the chosen topic; (2) there is no hard restriction on when to switch the topic; the model will determine an appropriate way to do the transition by itself. Such observations confirm that CP-VAE possesses a filled constrained latent space which make the latent code robust to manipulation across different time steps, which can be effectively reflected in the generation process. Due to space limitations, we put more samples in Appendix F.

## 7 Conclusion

In this work, we investigate latent vacancy as an important problem in unsupervised learning of controllable representations when modelling text with VAEs. To mitigate this, we propose to constrain the posterior within a learned probability simplex, achieving the first success towards controlled text generation without supervision.

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

## A    DETAILS ABOUT EXPLORATORY EXPERIMENTS

### A.1    MODEL DETAILS

For the $\beta$-VAE we used for the exploratory experiments, we use a LSTM encoding network and a LSTM decoding network. For the encoding network, the input size is 256, and the hidden size is 1,024. For the decoding network, the input size is 256, the hidden size is 1,024, and dropouts with probability 0.5 are applied on after the embedding layer and the LSTM layer in the decoding network. $\beta$ is chosen as 0.35, the dimension for the latent code is 80, and the batch size is 32. We use SGD with learning rate 1.0 to update the parameters for both the encoding and the decoding network. We train the model until the reconstruction loss stops decreasing.

### A.2    IDENTIFYING THE LATENT FACTOR INDICATING THE SENTIMENT

First, we normalize the value of each latent code by subtracting the mean estimated over all the training samples. Then we use the polarity of each latent code to classify the sentiment in the validation set. The one with the highest accuracy is identified as the latent factor indicating the sentiment.

### A.3    SAMPLES GENERATED FROM $\beta$-VAE

Table 5: Samples of generated sentences from $\beta$-VAE on Yelp.

|  | *Positive to Negative* | *Negative to Positive* |
|---|---|---|
| SRC | this place is super yummy ! | but it probably sucks too ! |
| $\beta$-VAE | this place is perfect for all of us or so long and over priced ! | thank you ! |
| SRC | i will be going back and enjoying this great place | there is definitely not enough room in that part of the venue . |
| $\beta$-VAE | i will be going back and recommending this place to anyone who lives in the valley ! | there is great . |

### A.4    MANIPULATION STRATEGIES

Following manipulation strategies have been attempted: (1) fixing the relevant factor to $\mu + 2\sigma$ and $\mu - 2\sigma$; (2) fixing the relevant factor to $\mu - \sigma$ and $\mu - \sigma$; (3) fixing the relevant factor to the maximum value and the minimum value of the relevant factor appearing in the training samples; (4) calculating a latent vector based on 10 manually constructed parallel sentences with opposite sentiment while keeping other factors unchanged. However, none of these four strategies is effective considering the generation results. We report the result with the first strategy in the paper, since it performs the best considering the accuracy and the BLEU score.

## B    DETAILS ABOUT EXPERIMENTS ON TEXT STYLE TRANSFER

### B.1    TRAINING REGIMES

Across all the datasets, we use Adam with learning rate 0.001 to update the parameters for the encoding network, while SGD with learning rate 1.0 to update the parameters for the decoding network. The batch size is chosen to be 32. Dropouts with drop probability 0.5 are applied on applied on after the embedding layer and the LSTM layer in the decoding network. We train the model until the reconstruction loss stops decreasing.

### B.2    MITIGATING POSTERIOR COLLAPSE

For the structured part $z^{(1)}$, we use $\beta$-VAE setting $\beta$ as 0.2 across all the datasets. For the unstructured part $z^{(2)}$, different strategies are employed for each dataset:

- **Yelp**: $\beta$-VAE setting $\beta$ as 0.35.
- **Amazon**: $\beta$-VAE setting $\beta$ as 0.35.
- **AG-News**: KL annealing, from 0.1 to 1.0 in 10 epochs.

## B.3 HYPERPARAMETER SETTINGS

Table 6: Hyperparameter settings.

|  | Yelp | Amazon | AG-News |
|---|---|---|---|
| Number of variations $K$ | 3 | 3 | 10 |
| Parameter to control the KL $\alpha$ | 100 | 100 | 10 |
| Input dimension for LSTM encoder | 256 | 256 | 512 |
| Hidden dimension for LSTM encoder | 1024 | 1024 | 1024 |
| Dimension for $\boldsymbol{z}^{(2)}$ | 64 | 64 | 96 |
| Dimension for $\boldsymbol{z}^{(1)}$ | 16 | 16 | 32 |
| Input dimension for LSTM decoder | 128 | 128 | 512 |
| Hidden dimension for LSTM decoder | 1024 | 1024 | 1024 |

The hyperparameters are chosen by checking $\mathcal{L}_{\text{VAE}}$, KL, and the generated outputs on the development set for **Yelp** and **AG-News**. **Amazon** follows the same setting as **Yelp** without extra tuning.

## B.4 DECODING STRATEGY

For decoding, we use beam search with a beam size of 5.

## C PROOF OF MINIMALIZATION OF EQ. 5

The problem can be formulated as an optimization problem as follows:

$$\text{maximize} \sum_{i=1}^{K} p_i^2, \quad \text{subject to} \sum_{i=1}^{K} p_i = 1.$$

By introducing a Lagrange multiplier $\lambda$, the Lagrange function is defined as

$$\mathcal{L}(p_1, p_2, \ldots, p_K, \lambda) = \sum_{i=1}^{K} p_i^2 - \lambda(\sum_{i=1}^{K} p_i - 1).$$

In order to find the optimal point, we require that

$$\frac{\partial}{\partial p_i} \left( \sum_{i=1}^{K} p_i^2 - \lambda(\sum_{i=1}^{K} p_i - 1) \right) = 2p_i - \lambda = 0, \quad i = 1, 2, \ldots, K,$$

which shows that all $p_i$ are equal. By using the constraint $\sum_i p_i = 1$, we find $p_i = \frac{1}{K}, i = 1, 2, \ldots, K$. By plugging into the results, $\boldsymbol{\mu}^{\top}\boldsymbol{\mu} = \alpha \sum_i p_i^2$ reaches its minimum $\frac{\alpha}{K}$.

## D COMPARISONS WITH BASELINES ON TOPIC MODELLING

**Experimental setup:** We use the AG news dataset for this task constructed by (Zhang et al., 2015). It contains four topic categories which are *World, Sports, Business* and *Sci/Tech*, with the title and description fields. For each category, there are $30,000$ training samples and $1,900$ test samples. In this paper, we drop the title and just use the description field. We compare our approach to two standard baselines for unsupervised topic modelling: (1) **LDA** (Blei et al., 2003), a standard implementation of LDA is used for this baseline[4]; (2) $k$-**means**. To show the power of our approach

---

[4]https://radimrehurek.com/gensim/

beyond the pre-trained sentence representations, we perform $k$-means clustering directly on the sentence representations. Following (Manning et al., 2010), we assign each inferred topic to one of the gold-standard topics with the optimal mapping and report the precision (*a.k.a.* purity), recall (*a.k.a.* collocation) and $F_1$ score. The number of topics is chosen to be 10. The results reported for the baselines and our model are the average over 10 runs.

**Quantitative results:**  The results are shown in Table 7. We can see that our approach achieves comparable results to **LDA** while significantly outperforming $k$-**means** in all four categories, indicating that our approach can go beyond just clustering on pre-trained sentence representations.

Table 7: Results for topic identification.

| Topic | Model | Precision | Recall | $F_1$ |
|---|---|---|---|---|
| World | LDA | 69.73 | **75.32** | 72.14 |
|  | $k$-means | 67.64 | 47.63 | 55.90 |
|  | Ours | **80.83** | 70.55 | **74.59** |
| Sports | LDA | 79.17 | 82.50 | **80.22** |
|  | $k$-means | 47.66 | **89.50** | 62.04 |
|  | Ours | **81.14** | 78.88 | 79.49 |
| Business | LDA | **72.10** | 66.45 | **68.46** |
|  | $k$-means | 53.06 | 53.16 | 53.11 |
|  | Ours | 64.04 | **64.53** | 63.97 |
| Sci/Tech | LDA | 66.55 | 59.77 | 61.60 |
|  | $k$-means | **81.32** | 31.59 | 44.67 |
|  | Ours | 65.20 | **71.74** | **66.77** |

# E   TEXT TRANSFER EXAMPLES

## E.1   SENTIMENT MANIPULATION ON YELP DATASET

Table 8: Sentiment manipulation results from positive to negative

| | |
|---|---|
| SRC | this was the best i have ever had ! |
| B-GST | this was the worst place i have ever had ! |
| **CP-G** | this was the worst pizza i have ever had ! |
| **CP-B** | this was the worst i have ever had ! |
| SRC | friendly and welcoming with a fun atmosphere and terrific food . |
| B-GST | the hummus is ridiculously bland and bland . |
| **CP-G** | rude and unorganized with a terrible atmosphere and coffee . |
| **CP-B** | the hummus is ridiculously greasy and tasteless . |
| SRC | i ordered the carne asada steak and it was cooked perfectly ! |
| B-GST | i ordered the carne asada steak and it was just as bad ! |
| **CP-G** | i ordered the carne asada steak and it was n't cooked and it was lacking . |
| **CP-B** | i ordered the carne asada burrito and it was mediocre . |
| SRC | the owner is a hoot and the facility is very accommodating . |
| B-GST | the owner is a jerk and the facility is very outdated . |
| **CP-G** | the owner is a hoot and the facility is empty and the layout is empty . |
| **CP-B** | the owner is a riot and the facility is very clean. |
| SRC | i will be going back and enjoying this great place ! |
| B-GST | i wo n't be going back and this place is horrible ! |
| **CP-G** | i will be going back and eat this pizza hut elsewhere . |
| **CP-B** | i will be going back and hated the worst dining experience . |

Table 9: Sentiment manipulation results from negative to positive

| | |
|---|---|
| SRC | there is definitely not enough room in that part of the venue . |
| B-GST | there is plenty enough seating in that part of the venue . |
| **CP-G** | there is definitely an authentic dinner in that part . |
| **CP-B** | there is definitely a nice theatre in that part . |
| SRC | but it probably sucks too ! |
| B-GST | but it tastes great too ! |
| **CP-G** | but it 's truly fun and insanely delicious . |
| **CP-B** | but it 's probably wonderful when u ! |
| SRC | always rude in their tone and always have shitty customer service ! |
| B-GST | always in tune with their tone and have great customer service . |
| **CP-G** | always great with their birthdays and always excellent music . |
| **CP-B** | always accommodating and my dog is always on family . |
| SRC | i was very sick the night after . |
| B-GST | i was very happy the night after . |
| **CP-G** | i was very pleased with the night . |
| **CP-B** | i was very happy with the night . |
| SRC | this is a horrible venue . |
| B-GST | this is a wonderful venue . |
| **CP-G** | this is a great place for celebrating friends . |
| **CP-B** | this is a great place for beginners . |

E.2   SENTIMENT MANIPULATION ON AMAZON DATASET

Table 10: Sentiment manipulation results from positive to negative

| SRC | most pizza wheels that i ve seen are much smaller . |
|---|---|
| B-GST | most pizza dough that i ve seen are much better . |
| **CP-G** | most pizza wheels that i ve seen are much more good and are much quality . |
| **CP-B** | most pizza wheels that i ve seen are much better than are much better |
| SRC | however , this is an example of how rosle got it right . |
| B-GST | however , this game is an example of how rosle loves it . |
| **CP-G** | however , this is an example of how toxic . . . sad . . . obviously . |
| **CP-B** | however , this is an example of how cheap . similar . cheap advice . cheap advice . similar . |
| SRC | auto shut off after num_num hours , which is a good feature . |
| B-GST | auto shuts off after num _ num hours , which is a shame . |
| **CP-G** | whipped mask off after num_num hours , which is slimy , which is disgusting . |
| **CP-B** | auto shut off after num_num hours, which is a stupid idea , which seems to be bad . |
| SRC | that said , the mic did pic up everything it could . |
| B-GST | that said, the game took up everything it could . |
| **CP-G** | that said, the shampoo did nt smell him well . stopped cleaning everything . ended up smelling sick |
| **CP-B** | that said, the mic did not fit everything on well , let me down it weren t cleaning |
| SRC | i also prefered tha blade weight and thickness of the wustof ! |
| B-GST | i also like the blade weight and of the wustof . |
| **CP-G** | i also disliked the blade weight and thickness of the materials . |
| **CP-B** | i also slammed the blade weight and thickness of the wide . |

Table 11: Sentiment manipulation results from negative to positive

| SRC | the quality is declined quickly by heat exposure . |
|---|---|
| B-GST | the water is quickly drained by head exposure . |
| **CP-G** | the quality is utilitarian so grinding or sandwiches . |
| **CP-B** | the quality is priceless quickly by heat rises . |
| SRC | the directions were easy to follow but the quality of the easel was pathetic . |
| B-GST | the directions were easy to follow but the quality of the product was excellent . |
| **CP-G** | the directions were easy to follow but the quality is good for the quality and is |
| **CP-B** | the directions were easy to follow but the quality is what the quality is like the best quality of |
| SRC | multiplayer is just as bad, though thankfully not worse . |
| B-GST | quality is just as good , though thankfully not perfect . |
| **CP-G** | besides it is just good , though . those . . usually . . . usually . . . |
| **CP-B** | multiplayer is just as bad, though somebody s also so far not so far but no problem . |
| SRC | another energy product that simply wastes our money . |
| B-GST | another energy product that simply saves our money . |
| **CP-G** | another energy product that simply glides your pasta . |
| **CP-B** | another energy product that simply wastes this money . |
| SRC | i received the wrong color and it shreds easily . |
| B-GST | i received the color and it works easily . |
| **CP-G** | i low the new color and it closes easily . |
| **CP-B** | i received the wrong color and it pours easily from dishwasher and dries easily on garlic easily . |

## F  TEXT TRANSITION EXAMPLES ON AG NEWS

Table 12: Topic transition examples.

| | |
|---|---|
| *World* throughout | BAGHDAD (Reuters) - Iraq 's interim prime minister , Iyad Allawi , said on Monday that the United States had no intention of withdrawing from the country to end the violence in Iraq . |
| *World* to *Sports* | BAGHDAD (Reuters) - Iraq 's interim prime minister , Iyad Allawi , said on Monday that the United States **had no intention of withdrawing its troops from the country to the end of the year .** |
| *World* to *Business* | BAGHDAD (Reuters) - Iraq 's interim prime minister , Iyad Allawi , said on Monday that the United States **had no intention of withdrawing its troops from the country to the country .** |
| *World* to *Sci/Tech* | BAGHDAD (Reuters) - Iraq 's interim prime minister , Iyad Allawi , said on Monday that the United States **had no intention of withdrawing its uranium enrichment program to the United States .** |
| *Sports* throughout | For the first time in four years , the US men 's basketball team won the gold medal in the men 's 400-meter medley relay . |
| *Sports* to *World* | For the first time in four years , the US men 's basketball team won the gold medal at **the Athens Olympics in Athens , where the United States and the United States have agreed to a peace deal .** |
| *Sports* to *Business* | For the first time in four years , the US men 's basketball team won the gold medal at **the Athens Olympics on Wednesday , with a surge in crude oil prices .** |
| *Sports* to *Sci/Tech* | For the first time in four years , the US men 's basketball team won the gold medal in **the men 's Olympic basketball tournament in Beijing on Tuesday .** |
| *Business* throughout | NEW YORK (Reuters) - U.S. stocks opened higher on Friday , as oil prices climbed above $48 a barrel and the Federal Reserve raised interest rates by a quarter percentage point . |
| *Business* to *World* | NEW YORK (Reuters) - U.S. stocks opened higher on Friday , as oil prices climbed above $48 a barrel **and the Federal Reserve raised interest rates by a quarter percentage point .** |
| *Business* to *Sports* | NEW YORK (Reuters) - U.S. stocks opened higher on Friday , as oil prices climbed above $48 a barrel **and the Federal Reserve raised interest rates by a quarter percentage point .** |
| *Business* to *Sci/Tech* | NEW YORK (Reuters) - U.S. stocks opened higher on Friday , as oil prices climbed above $48 a barrel **and the Federal Communications Commission said it would allow the companies to use mobile phones .** |
| *Sci/Tech* throughout | SINGAPORE (Reuters) - South Korea 's Hynix Semiconductor Inc. said on Tuesday it had developed a prototype micro fuel cell recharger for a range of security vulnerabilities in India . |
| *Sci/Tech* to *World* | SINGAPORE (Reuters) - South Korea 's Hynix Semiconductor Inc. said on Tuesday it had developed a prototype micro fuel **cell aimed at ending a standoff with North Korea .** |
| *Sci/Tech* to *Sports* | SINGAPORE (Reuters) - South Korea 's Hynix Semiconductor Inc. said on Tuesday it had developed a prototype micro fuel **cell aimed at protecting the world 's biggest gold medal .** |
| *Sci/Tech* to *Business* | SINGAPORE (Reuters) - South Korea 's Hynix Semiconductor Inc. said on Tuesday it had developed a prototype micro fuel **cell aimed at protecting the world 's largest oil producer .** |

