# OpenReview forum: "On Variational Learning of Controllable Representations for Text without Supervision"
_ICLR.cc/2020/Conference — Reject_

### Official Review · AnonReviewer2 · 2019-10-23
**Official Blind Review #2**

**Rating:** 8

**Review:**

This paper tries to pinpoint why sequence VAEs haven't worked well to provide disentangled representations. The authors posits that this happens due to the fact that perturbing the intermediate representation (codes) pushes them in regions which are not seen in training, and hence the model is not well equipped to perform well for those codes. To address this, they augment the VAE objective with terms to ensure that codes are present in a probability simplex and the entire simplex is uniformly filled by codes. I thought this was a very good paper. I found the observation very interesting, and the supporting experiments confirm the hypothesis. However, since I do not actively work in this area, I am not sure how exciting the result will be to other researchers.

There's some related work on controlled text generation.
https://arxiv.org/pdf/1811.00552.pdf
https://arxiv.org/pdf/1811.01135.pdf
It will be good to mention them and possibly compare and contrast with the above work.

**Experience Assessment:**

I have read many papers in this area.

**Review Assessment: Checking Correctness Of Derivations And Theory:**

I assessed the sensibility of the derivations and theory.

**Review Assessment: Checking Correctness Of Experiments:**

I assessed the sensibility of the experiments.

**Review Assessment: Thoroughness In Paper Reading:**

I read the paper thoroughly.

---

> ### Author Response · Authors · 2019-11-07
> **Thanks for your reviews! The mentioned related work has been added in the updated version.**
>
> Since our proposed model is trained without explicit labelled information unlike previous works trained in a supervised fashion (same for the two papers you mentioned), we just choose several representative works to restrain the content into 8 pages for the submitted version. We will have a more thorough discussion of related work in the camera-ready version.

---

### Official Review · AnonReviewer1 · 2019-10-23
**Official Blind Review #1**

**Rating:** 3

**Review:**

The paper "On Variational Learning of Controllable Representations for Text without Supervision" tackles the problem of latent vacancy of text representation via variational text auto-encoders. Based on the observation that a single factor of the sentence encoding gathers most of relevant information for classifying the sentence as positive or negative (sentiment classification), authors study the impact of manipulating this factor in term of the corresponding decoded sentences. They reasonnably claim that if such a manipulation fails at decoding accurate sentences, it is because we fall in representation areas that the decoder never seen during training. Thus they propose a way to constrain the posterior mean to a
learned probability simplex and only perform manipulation within the probability simplex.

The tackled problem is important. Variationnal text auto-encoding is a very challenging task, for which no perfect solution has been proposed yet. A important issue is that usually the posterior collapses, with the auto-regressive decoder eventually ignoring the codes. Another problem is that indeed the representation space can contain many holes, in codes have never been seen during training. The authors propose to cope with both problems by encouraging a part of the code mean to live in a simplex, which prevents from posterior collapsing. Next, to ensure that information is filled in this constrained part, they define a pairwise ranking loss which enforce the mean of each sentence to be more similar to the output of the encoding lstm than the mean of other sentences. For this part, more intuition justification is needed to well understand the effect of this additional loss. In what sense does it ensure that the space does not contain holes ? What ensures that the constrained part of the code is actually used by the decoder ?

My main concern is with regards to the experiments, which are clearly not enough detailled. First, I cannot understand what NLL is considered in the preliminary experiments. Authors study the effect of code manipulation on an NLL. But the likelihood of what ? Of the encoded sentence ? If yes it is natural that the NLL is impacted since we move the representation so the manipulated representation encodes another sentence... Or maybe it is w.r.t. a generated sentence from the obtained code ? But what sense does it make to assess the nll of the generated sentence ? Ok if the distribution is to flat, the NLL would not be good, but is it really what we want to observe ? Also, authors compare the impact of modifications on the representations of $\beta$-VAE with modifications on their model, but these are not the same modifications. What ensure that they have the same magnitude ? I cannot understand the paragraph on vp and vn in the experimental setup. Comparisons with metrics on text style transfer are also difficult to understand to me. What is the reference sentence ?

Minor questions:
    - z(1) is said to be parametrized by a MLP with sentences representations before eq2 and is said to be encoded by a LSTM after eq 2. What am I missing ?
     - Please better detail fig2



**Experience Assessment:**

I have published one or two papers in this area.

**Review Assessment: Checking Correctness Of Derivations And Theory:**

I carefully checked the derivations and theory.

**Review Assessment: Checking Correctness Of Experiments:**

I assessed the sensibility of the experiments.

**Review Assessment: Thoroughness In Paper Reading:**

I read the paper at least twice and used my best judgement in assessing the paper.

---

> ### Author Response · Authors · 2019-11-08
> **Thanks for your reviews! Your concerns are addressed in the comment.**
>
>
> - Response to the concern about the preliminary experiments:
>
> As described in the paper, "we empirically estimate the density of sentiment-manipulated codes under the aggregated posterior distribution of our trained VAE". Formally, it is the averaged NLL under the aggregated posterior distribution, which is $q_{\phi}(z) = \mathbf{E}_{x \sim p(x)} q_{\phi}(z|x)$ defined in Sec. 2. In other words, it is not the NLL of sentences (neither original nor generated). The reason for showing the density of manipulated code under aggregated posterior to support the “latent vacancy” hypothesis, i.e. manipulated codes land in region where decoder is not trained for. For this purpose, Fig 3 illustrates both the problem as well as the effect of our fix.
>
> In terms of fairness of comparison, the goal of the preliminary experiments is to validate our “latent vacancy” hypothesis about the aggregated posterior, instead of comparing the absolute values of NLL of the codes. Furthermore, although $\beta$-VAE and CP-VAE are different considering encoder architecture and loss function, they both have 80-dimensional latent codes in the preliminary experiments as mentioned in the paper.
>
> - Response to the concern about the experimental setup:
>
> In experimental setup, it should be “To decide which basis vector corresponds to which sentiment, we sample 10 positive and 10 negative sentences respectively in the development set, pass them to the encoder, and choose the basis vector with the highest average p_i,  yielding v_p as the positive basis and v_n as the negative basis.” We clarify it in the updated version.
>
> - Response to the concern about the results on text style transfer:
>
> Generally, the experiments follow the evaluation protocols in (Radford et al., 2019) which is the current SOTA method on text style transfer. The reference sentence is used for calculating GLUE score which has a better correlation with the human judgement argued by (Radford et al., 2019). We include these reference sentences in the updated version.
>
> Q: In what sense does our model ensure that the space does not contain holes?
>
> As we can see from the Figure 3, the aggregated posterior distribution of our proposed model is relatively stable in the constrained space, which indicates that each point in this space possess a relative high likelihood. It’s also verified by the fact the NLL of the original and the modified codes are largely overlapped. In addition, Figure 4 shows that L_{S-REC} can help codes of all the observed training samples spread evenly in our learnt probability simplex, which validates our intuition. All these observations validate that our proposed model can actually help alleviate the latent vacancy problem observed in this paper.
>
> Q: What ensures that the constrained part of the code is actually used by the decoder?
>
> As mentioned in the Sec. 4.2, the KL term for $z^{(1)}$ will never collapse due to the structure of our constrained posterior space. Due to this irradicable penalty term, the decoder is forced to learn something useful for this part of the code. For the experiments conducted, we set alpha as a quite large value to maintain the magnitude of the KL term. The hyperparameter settings can be found in Appendix B.3.
>
> - Response to minor questions:
>
> 1. $z^{(1)}$ is parametrized by a MLP with sentence representations as stated in the Sec. 4.1. There is a typo in the Sec. 4.2 which should be “MLP” instead of “LSTM”. This typo has been corrected in the updated version.
>
> 2. Clarification on Figure 2: Sorry for the poorly organized figure. We have changed Figure 2 and added reference to better explain the figure in the updated version. Generally, the purpose of Figure 2 is to illustrate how we address the issue depicted in Fig 1. Our proposed approach is to constrain the posterior in a way that the manipulation only happens in a learned simplex. The green double arrow indicates the way we perform the manipulation matching the green curve plotted in Figure 3(a).

---

### Official Review · AnonReviewer3 · 2019-11-02
**Official Blind Review #3**

**Rating:** 6

**Review:**

This paper presents a method for controlled text generation by using a new loss function (standard VAE loss with auxiliary losses added on). The method is tested on style transfer datasets: Yelp and Amazon. The central hypothesis is that when manipulating latent codes of a VAE, you can end up in low-density regions of the aggregated posterior. Such latent codes are rarely seen by the decoder so that quality of generation is low. To address this problem, they constrain the posterior mean to a learnt probability simplex and try to ensure that the simplex is densely filled. They do this by adding 2 regularizing losses to the VAE loss.

Some questions:-
- I had some trouble following section 4.2 where the derivation for L-reg is explained, this is the loss that constrains the posterior to a simplex. Is E = [e_1, ..., e_k] learnt or a hyperparameter? Most of the text seems to indicate that it is learnt, but you mention that alpha is a tunable hyperparameter, where alpha = e_i^2. Could you elaborate on how e_i^2 can be fixed to alpha is E is learnt?
- You state that mu^2 reaches a minimum at alpha/K. Could you please provide a proof? This could be placed in an Appendix.
- In section 6.3, is K still 3 or is it now >=4?

Some citations issues
- You're missing a couple key citations. I think your work should cite Hu et al.'s (2018) Toward Controlled Generation of Text.
- You should definitely cite Lample et al.'s (2019)  Multiple-Attribute Text Rewriting. The results in Lample et al. are definitely comparable, and often better than the results presented in this paper. While I think this paper makes an independent and useful contribution, some of the claims are overblown since the results from Lample et al. are not presented or discussed. For example, the Lample's model on Yelp achieves a higher accuracy and a lower perplexity than the CP-VAE model presented in this paper.
- Correction of citations: Kim et al. (2018), Adversarially regularized autoencoders, is in fact Zhao et al. (2018), you're just missing the first author.


Additionally, I'd like to see some more detail about the robustness of the method. Is the model robust to initialization and hyperparameter settings? Is posterior collapse always avoided? In Table 1 you show that without L-Reg the CP-VAE suffer with posterior collapse, but this would carry more weight if you showed some statistics on how often using L-Reg helps avoid posterior collapse. In general it looks like we're mostly presented with the best results, I'd like to know a little more about the mean and standard deviation of the model runs as well.

Overall, I think this paper makes a worthwhile contribution. I was unclear on a few parts of the derivation for the regularizing losses, but the intuition seems sensible. The presented results are impressive but have a few missing pieces: citing some key results (Lample et al. 2019) and including results about robustness. I think the questions I have, and any shortcomings this paper has, could be addressed in a camera-ready version.


Other bits,
- I do not understand figure 2. It's also never referred back to once we learn what v_p and v_n are. Some clarification here would be helpful.
- Section 5.1, line 3: achieve -> achieved

**Experience Assessment:**

I have read many papers in this area.

**Review Assessment: Checking Correctness Of Derivations And Theory:**

I assessed the sensibility of the derivations and theory.

**Review Assessment: Checking Correctness Of Experiments:**

I assessed the sensibility of the experiments.

**Review Assessment: Thoroughness In Paper Reading:**

I read the paper thoroughly.

---

> ### Author Response · Authors · 2019-11-07
> **Thanks for your reviews! Your concerns are addressed in the comment.**
>
> Q: Is E learnt or a hyperparameter?
>
> Yes, it is learnt. Sorry for the ambiguity in the original text. Actually, when L_{REG} is 0, $e_i^\top e_i = \alpha$. In this case, Eq. 5 reaches its minimum $\alpha/K$ when p is a uniform distribution. In practice, L_{REG} will quickly decrease to around 0, ensuring that the KL term will never fully collapse. We reword the original text and change the notations in the updated version.
>
> Q: Proof of mu^2 reaches a minimum at alpha / K.
>
> Provided in Appendix C in the update version.
>
> Q: In section 6.3, is K still 3 or is it now >=4?
>
> In section 6.3, K is set to be 10. The updated version contains it in the main text. The detailed hyperparameter settings can be found in Appendix B.3.
>
> - Results about Robustness
>
> From our observations, CP-VAE is not that sensitive to initialization and hyperparameter settings. Tuned hyperparameter settings help but the model will always learn disentangled representations and generate reasonable outputs with different hyperparameter settings. In practice, I just tune alpha (in Eq. 6) and beta (in $\beta$-VAE) to trade off between attribute control and content preservation, with other hyperparameters fixed.
>
> Considering posterior collapse, we restrain the posterior in a probability simplex and show that the KL term will never collapse if the learnt basis vectors are orthogonal to each other. Encouraged by the L_REG term, the learnt basis vectors can achieve orthogonality almost perfectly in a few training steps, which means that L_REG is very close to 0. As a result, the KL term will indeed never collapse in our experiments.
>
> We are running additional experiments and will update the results on robustness of our proposed model later in the rebuttal period.
>
> - Citation Issues
>
> The mentioned related work has been added in the updated version. For the concern on comparisons with Lample et al. (2019), their proposed methods are still dependent on labelled information like the other supervised methods we compare in Sec. 6.2. Our experiments on text style transfer is not to show that our proposed model is superior than supervised  models, but rather to validate our assumption about why traditional VAEs work not that well on this task in an unsupervised setting. Empirically, we find that our unsupervised approach can achieve results comparable with previous supervised approaches considering metrics on text style transfer. For the submitted version, we just choose several representative works to restrain the content into 8 pages. We will have a more thorough discussion of related work in the camera-ready version.
>
> - Clarification on Figure 2
>
> Sorry for the poorly organized figure. We have changed Figure 2 and added reference to better explain the figure in the updated version. Generally, the purpose of Figure 2 is to illustrate how we address the issue depicted in Fig 1. Our proposed approach is to constrain the posterior in a way that the manipulation only happens in a learned simplex. The green double arrow indicates the way we perform the manipulation matching the green curve plotted in Figure 3(a).

---

### Public Comment · ~Ali_Razavi1 · 2019-11-06
**Missing prior works citation**

I would like to mention that one of the main ideas proposed in this paper, that is imposing hard constrains on the posterior to prevent the posterior collapse problem was explored in our work on Delta-VAE published in ICLR 2019:
https://openreview.net/forum?id=BJe0Gn0cY7

We presented two different ways of achieving this (first is using AR(1) prior for sequential latent variables and the second is independent delta-vae for diagonal gaussians presented in Appendix D).

We had experiments on both text and image datasets, and presented evidence of controllable text generation in our samples (see Figure 12, 13 and 14) in the appendix. The authors can find several other prior publications that should be cited in the related work section of our paper.

---

> ### Author Response · Authors · 2019-11-08
> **Thanks for your comments. We have added the suggested citation in the related work of the updated version.**
>
> Since our primary goal in this paper is to mitigate the latent vacancy problem observed instead of just preventing the posterior collapse, we just choose several representative works to restrain the content into 8 pages for the submitted version. But we agree that our constraint does force an explicit gap in the KL, we will have a more thorough discussion of related work in the camera-ready version.
>
> Another thing we would like to clarify is that imposing hard constraints alone is not our main contribution. Rather, it’s about observing and validating the “latent vacancy” hypothesis. This directly motivates what kind of hard constraints makes controlled text generation easier.
>
> Finally, we see three sets of samples from the latent space interpolation qualitative study in the appendix of the delta-vae work. While those are interesting, we don’t think they are comparable to the ones presented in our work, because the relevant factors (e.g., sentiment, topic) are not identified and controlled, and there is no quantitative evaluation.

---

### Author Response · Authors · 2019-11-12
**Updated version uploaded.**

Dear reviewers and all,

Thanks for the constructive comments from all the reviewers. An updated version has been uploaded considering all the reviewers’ concerns.  We believe this version is much clearer in written with less ambiguity, which would not be possible without all the useful feedback from the reviewers. We also appreciate any further questions and concerns about this work.

The main changes are summarized as follows:

1. We updated Table 1 to include results on robustness of our proposed model as pointed out by R3. The latest results include the mean and standard deviation calculated with 5 different runs. As we can see from the updated results, our proposed model can consistently produce close results reported in the original version, with reasonable variance.

2. We simplified Figure 2 and added reference in Sec. 3.2 to better explain the figure, following the advice from R1 and R3.

3. We provided the proof of minimization of Eq.5 in Appendix C, as required by R3.

4. We changed the notations in Eq. 4 and Eq. 5 to avoid ambiguity and reworded the text around these two equations to better deliver our meaning, as mentioned by R3.

5. We corrected several typos pointed out by R1 and R3.

6. We added the citations in the related work section, pointed by the reviewers and other public comments.

---

### Author Response · Authors · 2019-12-23
**Response to the meta-review.**

It is unfortunate that our work is rejected. We understand that the decision is final and appealing is not possible. However, we would like to address some misunderstanding of meta-reviewer and R1, for future readers’ sake.

First, our method actually outperforms many supervised approaches in text style transfer, as shown in our experiments. Considering the unsupervised nature of our method, it is already impressive to achieve comparable results with the current SOTA supervised approaches. Let alone the fact that the current SOTA leverages pre-trained GPT2 for generation, while our proposed method train the generator from scratch.

Second, as we have said in the paper, and have repeated in the rebuttal,  “The goal of the preliminary experiments is to validate our ‘latent vacancy’ hypothesis about the aggregated posterior, instead of comparing the absolute values of NLL of the codes.” The fact that NLL of manipulated code under aggregated posterior is less affected in our model is precisely the reason why ours work better. As for whether the magnitude of manipulation is comparable, we manipulated the code to the edge of the constraint space in our model, the maximum allowed range. Furthermore, as Fig 3 b) and c) shows, it’s not just about the magnitude of manipulation, but also the fact that the density of manipulated code under beta-VAE exhibit multiple modes, while just a single mode in our model which closely match original code, clearly confirming the latent vacancy hypothesis.

Last but not least, “some of the generated samples are not very convincing” is vague and subjective reason to reject a paper. If there is another unsupervised system that can produce sentences as natural as ours with fine-grain control over topics that allow switching within a sentence, with or without expensive large scale pre-training like GPT-2, we welcome suggestions for comparison.

---

### Decision · Program_Chairs · 2019-12-19

**Decision:**

Reject

**Comment:**

This paper analyzes the behavior of VAE for learning controllable text representations and uses this insight to introduce a method to constrain the posterior space by introducing a regularization term and a structured reconstruction term to the standard VAE loss. Experiments show the proposed method improves over unsupervised baselines, although it still underperforms supervised approaches in text style transfer.

The paper had some issues with presentation, as pointed out by R1 and R3. In addition, it missed citations to many prior work. Some of these issues had been addressed after the rebuttal, but I still think it needs to be more self contained (e.g., include details of evaluation protocols in the appendix, instead of citing another paper).

In an internal discussion, R1 still has some concerns regarding whether the negative log likelihood is less affected by manipulations in the constrained space compared to beta-VAE. In particular, the concern is about whether the magnitude of the manipulation is comparable across models, which is also shared by R3. R1 also think some of the generated samples are not very convincing.

This is a borderline paper with some interesting insights that tackles an important problem. However, due to its shortcoming in the current state, I recommend to reject the paper.